

# Assessment of inter-rater and intra-rater reliability of the Luna EMG robot as a tool for assessing upper limb proprioception in patients with stroke—a prospective observational study

Justyna Leszczak[1], Bogumiła Pniak[1,2], Mariusz Drużbicki[1], Anna Poświata[3], Michał Mikulski[3], Anna Roksela[3,4] and Agnieszka Guzik[1]

[1] Medical College, Institute of Health Sciences, University of Rzeszow, Rzeszów, Poland
[2] Excelsior Health and Rehabilitation Hospital, Iwonicz Zdrój, Poland
[3] EGZOTech Sp. z o.o., Gliwice, Poland
[4] Faculty of Automatic Control, Electronics, and Computer Science, PhD School, Silesian University of Technology, Gliwice, Poland

Corresponding author
Justyna Leszczak,
jleszczak@ur.edu.pl

## ABSTRACT

**Background:** The aim of the study was to assess the inter-rater and intra-rater agreement of measurements performed with the Luna EMG (electromyography) multifunctional robot, a tool for evaluation of upper limb proprioception in individuals with stroke.

**Methods:** The study was conducted in a group of patients with chronic stroke. A total of 126 patients participated in the study, including 78 women and 48 men, on average aged nearly 60 years (mean = 59.9). Proprioception measurements were performed using the Luna EMG diagnostic and rehabilitation robot to assess the left and right upper limbs. The examinations were conducted by two raters, twice, two weeks apart. The results were compared between the raters and the examinations.

**Results:** High consistency of the measurements performed for the right and the left hand was reflected by the interclass correlation coefficients (0.996–0.998 and 0.994–0.999, respectively) and by Pearson's linear correlation which was very high (r = 1.00) in all the cases for the right and the left hand in both the inter-rater and intra-rater agreement analyses.

**Conclusions:** Measurements performed by the Luna EMG diagnostic and rehabilitation robot demonstrate high inter-rater and intra-rater agreement in the assessment of upper limb proprioception in patients with chronic stroke.
The findings show that Luna EMG is a reliable tool enabling effective evaluation of upper limb proprioception post-stroke.

# INTRODUCTION

Proprioception is one of the senses making it possible to determine the position of one's body in space. It is an important part of the process involved in balance control, and in

sensing movement without visual control. Its development is impacted by learning and memory. Proprioception provides us with data about the position of individual joints or limbs and the degree of muscle tension in the specific region. Furthermore, owing to mechanoreceptors, information is transmitted to the central nervous system (*Korpanty et al., 2019*; *Valori et al., 2020*). Proprioception is important during the performance of functional movements and activities of daily living, and during rehabilitation (*Han et al., 2016*).

An important objective of stroke rehabilitation is to restore the functional capacity of the upper limb. Impaired proprioception significantly reduces the patients' ability to use the upper limb after stroke during activities of daily living. This results in their functional dependence and disability (*Doyle et al., 2010*). Patients with stroke have impairments which adversely affect their performance in motor tasks requiring sensorimotor information. Moreover, possible secondary complications, including injuries or wounds in the upper limb, may further reduce the patient's potential motor capacities and impair their image of their environment (*Doyle et al., 2010*). As an example, the affected hand adversely impacts the ability of patients with stroke to perform activities of daily living, such as driving a car or using a computer. In people after a stroke, not only does impairment of sensation occur, but it may even be lost, which is associated with adverse consequences of the functioning of the person after a stroke. The effects of loss of sensation on a patient after stroke include worse results from rehabilitation, lower level of functioning, lower quality of life, and they may have an impact on prolonging the patient's stay in hospital (*Patel et al., 2000*; *Sommerfeld & von Arbin, 2004*; *Kenzie et al., 2024*). Recent studies show that proprioception disorders occur in about 30 to 60% of people after a stroke (*Tulimieri & Semrau, 2024*).

The treatment and rehabilitation of the affected hand is a lengthy process and does not guarantee full functional recovery. Despite various problems, a great deal of effort is made to enhance the process of restoring the function of the upper limb (*Hernández et al., 2019*).

Many neurological diseases lead to impairment of proprioception, which is considered an important signal for training to induce neural reorganisation. Despite the widely recognised importance of proprioception in motor control and rehabilitation, there is no single method enabling a precise and objective assessment. It is important to assess both proprioceptive sensitivity, *i.e.*, the smallest detectable stimulus, and proprioceptive acuity, *i.e.*, the smallest perceptible difference between two detected stimuli. Proprioception assessment in practice involves sensory testing of limb position and joint position adjustment without visual input from the patient. These tests are not as accurate as required during rehabilitation. Therefore, researchers use specialised equipment that tests joint positions in a more accurate way. Such robotic devices have the ability to control passive movement of the limb (*Elangovan, Herrmann & Konczak, 2014*, *Han et al., 2016*).

Novel methods and techniques are mainly intended to enable patients with stroke to improve their functioning within the society. Today, functional therapy is frequently designed to apply a biofeedback method, with the use of robotic devices (*Kołcz-Trzęsicka et al., 2017*; *Kwolek et al., 2013*). One of such devices, applied in rehabilitation of the affected upper limbs, is LUNA EMG. In its operation it uses reactive electromyography to

train the sensorimotor cortex (*Olczak & Truszczyńska-Baszak, 2021*; *Zasadzka, Tobis & Trzmiel, 2020*).

In the literature there are few reports focusing on the use of the Luna EMG robotic device in assessment and rehabilitation of upper extremity (*Clinical Trials, 2024*). One study investigated inter-rater and intra-rater reliability of Luna EMG in the assessment of upper limb proprioception; however, it was conducted in a group of healthy individuals (*Leszczak et al., 2024*). Therefore the current study was designed to examine inter-rater and intra-rater reliability of the Luna EMG multifunction robotic device as a tool intended to assess upper limb proprioception in patients with a stroke.

## MATERIALS AND METHODS

### Type of study

The research was conducted in the form of a prospective observational study.

It was carried out in accordance with the ethical rules of the Declaration of Helsinki, and approved by the Local Bioethics Commission of the University of Rzeszow (resolution no. 2022/036/W). Written consent was obtained from all the participants. The study was registered in the clinical trials register at ClinicalTrials.gov (registration number NCT05486052).

### Participants

The minimum sample size was determined prior to the study, using a sample size calculator ("PLUS module" from Statistica 13.3 software). The procedure identified a population size of 77 patients with chronic stroke. Ultimately, 126 patients with chronic stroke were enrolled for the study.

The study was conducted in a rehabilitation and spa therapy hospital (physiotherapy laboratory) in the Podkarpackie Region, Poland. The inclusion criteria were defined as follows: completed first ischemic stroke; patient's informed, voluntary consent to participate; elementary (basic) gripping ability; the upper limb and hand paresis rated 4–5 on the Brunnström scale; degree of disability Rankin score of three; spastic tension of the upper limb and hand paresis not more than three on the modified Ashworth scale; and current health condition, confirmed by a medical examination, allowing participation in tests and exercises. The following exclusion criteria were applied: lack of the patient's informed, voluntary consent; second or subsequent ischemic stroke of the brain; haemorrhagic stroke; stroke of the brainstem and cerebellum; cerebellum disorders of the higher mental functions limiting the ability to understand and perform tasks during examinations; visual disturbances; mechanical and thermal injuries potentially impairing the hand grip function; concomitant neurological, rheumatological and orthopaedic diseases, including neuropathies that can affect proprioception, permanent contractures that may affect the grasping ability and locomotion; unstable medical condition (*i.e.*, cardiopulmonary disturbances, consciousness disturbances; convulsions, sudden acute chest pain, cardiac arrhythmias, dyspnoea); metal implants; electronic implants; menstruation in women and epilepsy.

The following inclusion and exclusion criteria were used based on available data in the literature (*Rinderknecht et al., 2018*; *Kenzie et al., 2024*; *Tulimieri & Semrau, 2024*; *Findlater et al., 2018*; *Morreale et al., 2016*; *Cherpin et al., 2019*; *Andersen et al., 2009*).

There were 200 patients with stroke in the rehabilitation and spa therapy hospital at the time of the study. Based on the inclusion and exclusion criteria, 147 patients were initially qualified to participate; however, after medical examination eight individuals were excluded. Ultimately 130 patients took part in Examination I, and 126 patients participated in Examination II (Fig. 1).

The study group comprised 126 individuals, including 78 women (61.9%) and 48 men (38.1%). The youngest participant was aged 45 *(Min. 45.0)*, and the oldest 75 years *(Max. 75.0)*, the mean age being nearly 60 years $(\bar{x} = 59.9)$. The respective mean values of body weight (in kilograms), body height (in centimetres) and body mass index (BMI) were 68.3 kg $(\bar{x} = 68.3)$, 170 cm $(\bar{x} = 170.0)$ and 23.6 kg/m$^2$ (Table 1).

## Procedure

The examinations were carried out using the diagnostic and rehabilitation device Luna EMG (EGZOTech Sp.z oo Gliwice, Poland). The assessments were carried out by two independent raters, under the same conditions and during the same time period. Two examinations of proprioception in relation to upper limb function were performed two weeks apart. The independent raters were physiotherapists with more than 5 years of experience working with people after stroke, trained in assessment, who had participated in previous published studies related to evaluation of the reliability of the Luna EMG Rehabilitation Robot to assess proprioception in the upper limbs in 102 healthy adults (*Leszczak et al., 2024*).

Luna EMG is a diagnostic and rehabilitation robotic device and its operation is based on reactive electromyography designed to train the sensorimotor cortex. The bioelectrical signals (EMG) obtained from the patient's muscles show that the movement is active (*Rinderknecht et al., 2018*; *Oleksy et al., 2022*; *EGZOTECH, 2019*).

Calibration of the Luna EMG device is performed annually by a trained service technician. This process includes a thorough validation of the measurement functions to ensure the accuracy and reliability of the device's readings. Regular calibration is essential for maintaining the device's performance and ensuring the consistency of test results over time. Before the start of the examination, the Luna EMG measured the weight of the limb to unweight it during the test procedure, thereby eliminating its impact on the patient's results. During the measurements, the patients remained in a sitting position, with the upper limb from the shoulder joint aligned with the trunk, and extended in the elbow, hips flexed at 90°. The trunk and the assessed limb were stabilised using straps. Each measurement comprised four tests performed in succession in each upper extremity. The range of motion was set between 0 and 90 degrees of elbow extension, which is limited by the device. The initial elbow joint position was 0° and the target position of elbow flexion was 60°; the movement was performed either actively or passively (*Leszczak et al., 2024*) (Fig. 2).

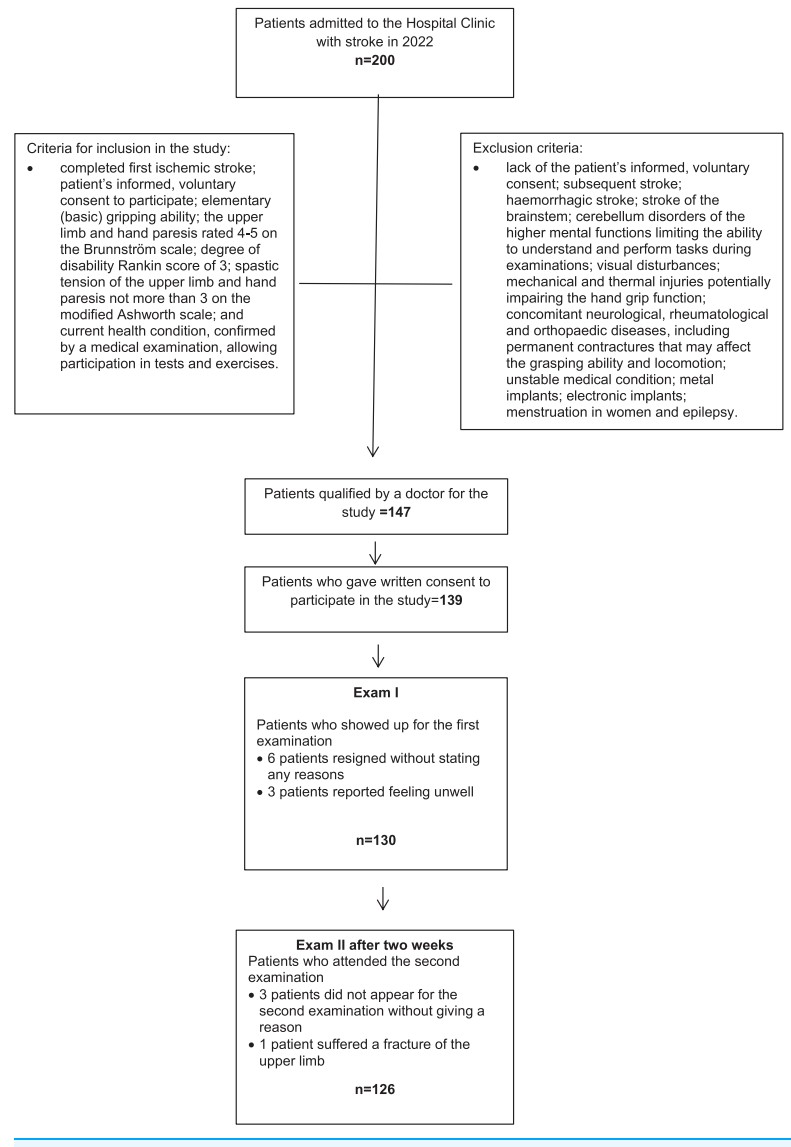

**Figure 1 Flow diagram.**

During the passive movement phase, the speed of the device was set to 30 degrees per second for the learning process and 10 degrees per second for evaluating the target position. The maximal torque for the device was maintained at 10 Nm, and the holding time during the learning phase of the target position (elbow flexion 60°) was set to 5 s.

Initially, the device passively conducted the patient's movement, then the patient was asked to "remember the movement/setting" during which time the equipment stopped for 5 s and the patient was tasked with learning to sense their position. Then, after 5 s, the limb passively returned to its initial position (elbow extension 0°). The initial position was held for 3 s. After learning a passive movement, the patient was asked to actively perform the same movement with a stop: on the request, "please move your elbow to the position you learned and hold your arm in that position." All tests were performed with the patient's eyes closed. The duration of the study of one patient by two researchers was 30 min.

**Table 1 Descriptive statistics–sex, age, body mass (kg), body height (cm) and body mass index (BMI).**

| Variables | | N | $n$ cumulative | Percent | Percent cumulative |
|---|---|---|---|---|---|
| Gender | Woman | 78 | 78 | 61.90 | 61.9 |
| | Man | 48 | 126 | 38.10 | 100 |
| Current place of residence | Urban area >100,000 population | 17 | 17 | 13.49 | 13.5 |
| | Urban area 50–100,000 population | 51 | 68 | 40.48 | 54.0 |
| | Urban area <50,000 population | 36 | 104 | 28.57 | 82.5 |
| | Rural area | 22 | 126 | 17.46 | 100 |
| Way of living | With husband/wife and children | 24 | 24 | 19.05 | 19.0 |
| | With husband/wife | 58 | 82 | 46.03 | 65.1 |
| | With children | 22 | 104 | 17.46 | 82.5 |
| | With another person | 16 | 120 | 12.70 | 95.2 |
| | Alone | 6 | 126 | 4.76 | 100 |
| Education | Primary | 12 | 12 | 9.52 | 9.5 |
| | Vocational | 39 | 51 | 30.95 | 40.5 |
| | Secondary | 48 | 99 | 38.10 | 78.6 |
| | Higher | 27 | 126 | 21.43 | 100 |
| Hypertension | Yes | 71 | 71 | 56.35 | 56.3 |
| | No | 55 | 126 | 43.65 | 100 |

| Variables | Descriptive statistics | | | | | | | |
|---|---|---|---|---|---|---|---|---|
| | N | $\bar{x}$ | Me | Min. | Max. | Q1 | Q3 | SD |
| Age | 126 | 59.9 | 59.0 | 45.0 | 75.0 | 55.0 | 66.0 | 7.46 |
| Body mass (kg) | 126 | 68.3 | 68.0 | 48.0 | 101.0 | 57.0 | 77.0 | 12.47 |
| Body height (cm) | 126 | 170 | 170 | 158 | 187 | 166 | 175 | 6.70 |
| BMI | 126 | 23.6 | 23.7 | 17.6 | 32.8 | 20.8 | 25.7 | 3.49 |
| Time since stroke (months) | 126 | 21.1 | 18.0 | 7.0 | 52.0 | 12.0 | 28.0 | 11.52 |

**Note:**
N, number of observations; $\bar{x}$, mean; Me, median; Min, minimum value; Max, maximum value; Q1, lower quartile; Q3, upper quartile; SD, standard deviation.

## Statistical analysis

The analysis was carried out in the Statistica 13.3 program and the JASP program 0.18.3.0 (*JASP Team (2024)*, Computer Software, Netherlands).

The mean and standard deviation values were first calculated for each series of measurements, and then for the differences between the series of measurements compared in the study. The significance of differences in the average level of the two measurement series was assessed using the t-test for dependent samples in which no significant values should be observed, but it should be noted that this is not a key factor in assessing the agreement of measurements. A comparative analysis of the measurement series was performed using Pearson's linear correlation coefficient, as well as the key measure of agreement between two measurements, *i.e.*, intraclass correlation coefficient (ICC). Bland-Altman method was applied as an alternative measure of agreement.

The value of $p < 0.05$ was assumed to reflect statistical significance.

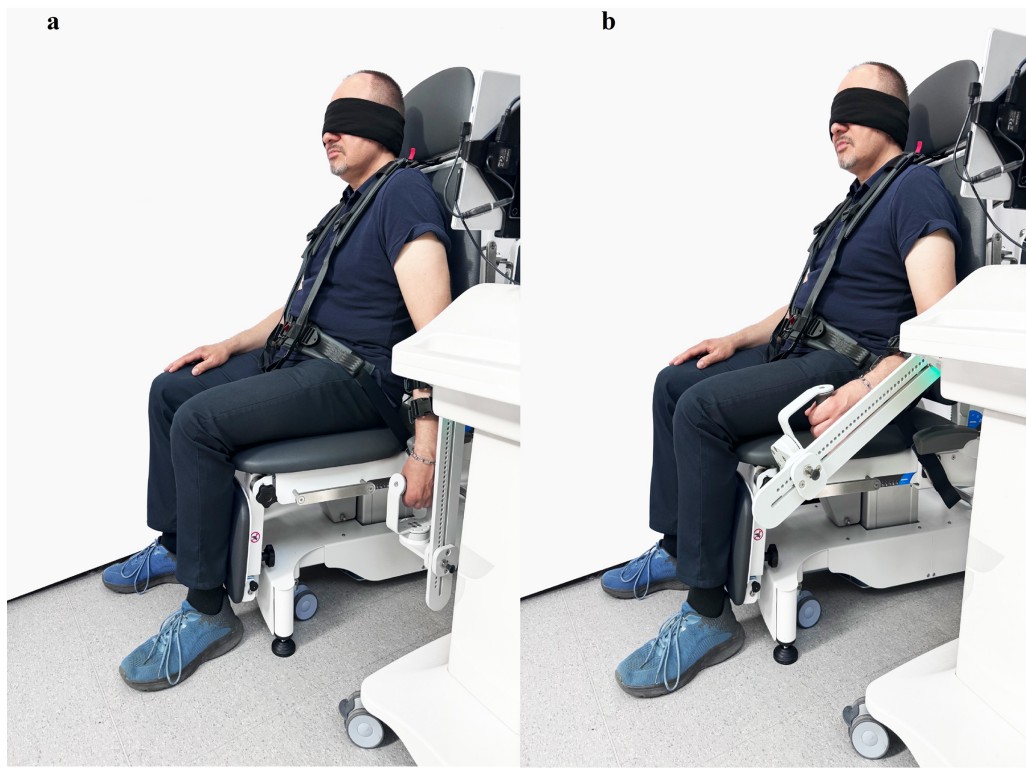

**Figure 2 Measurement of the sense of elbow flexion: (A) baseline position; (B) target position.**

## RESULTS

### Agreement between the measurements of right hand proprioception

As regards the assessment of the significance of differences, it was observed that Rater I on average recorded slightly higher measurement values in Exam I compared to those recorded in Exam II ($p$ = 0.0001). A similar relationship was found between the measurements performed by the Raters in Exam I; on average the values recorded by Rater I were significantly higher compared to those identified by Rater II ($p$ = 0.0220).

The above observations were not confirmed by Pearson's linear correlation coefficients, which in all the analyses assessing both inter-rater and intra-rater agreement were very high *(r = 1.00)*. The high agreement of the measurements carried out was also confirmed by intraclass correlation coefficients (ICC) (0.996–0.998) (Table 2).

The above findings are also confirmed by Bland-Altman plots. The highest mean deviation between measurements was identified in the assessment of agreement between Exam I and Exam II performed by Rater I (0.09°), and in a considerable majority of the results the differences were up to 0.4–0.5° (Fig. 3).

### Agreement between the measurements of left hand proprioception

In the assessments of left hand proprioception the findings show no significant differences between the mean results recorded by the two Raters ($p$ = 0.4487; $p$ = 0.4027); however, the mean results recorded by each Rater in the two Exams differed significantly ($p$ = 0.0002;

**Table 2 Agreement between measurements of right hand proprioception.**

**Proprioception (right hand) [°]**

| Exam | Rater | Measurement | | Differences | | p | r | ICC | CV | SEM |
|---|---|---|---|---|---|---|---|---|---|---|
| | | $\bar{x}$ | SD | $\bar{x}$ | SD | | | | | |
| I | 1 | 5.80 | 3.22 | −0.04 | 0.18 | 0.0220 | 0.998 | 0.998 | 55.54 | 0.29 |
| I | 2 | 5.77 | 3.18 | | | ($d = 0.207$) | | (0.998–0.999) | 55.19 | 0.28 |
| II | 1 | 5.72 | 3.15 | 0.01 | 0.20 | 0.7921 | 0.998 | 0.998 | 55.18 | 0.28 |
| II | 2 | 5.72 | 3.18 | | | ($d = -0.024$) | | (0.997–0.999) | 55.56 | 0.28 |
| I | 1 | 5.80 | 3.22 | −0.09 | 0.23 | 0.0001 | 0.998 | 0.997 | 55.54 | 0.29 |
| II | 1 | 5.72 | 3.15 | | | ($d = 0.373$) | | (0.996–0.998) | 55.18 | 0.28 |
| I | 2 | 5.77 | 3.18 | −0.05 | 0.27 | 0.0578 | 0.997 | 0.996 | 55.19 | 0.28 |
| II | 2 | 5.72 | 3.18 | | | ($d = 0.171$) | | (0.995–0.998) | 55.56 | 0.28 |

**Note:**
$\bar{x}$, mean; SD, standard deviation; °, degrees of angle; p, assessment of the significance of differences in the average level of two series of measurements (t-test for related samples); d, wielkość effect size; d, Cohena; r, Pearson's linear correlation coefficient; ICC, intraclass correlation coefficient with 95% confidence interval; CV, coefficient of variation; SEM, standard error of the mean.

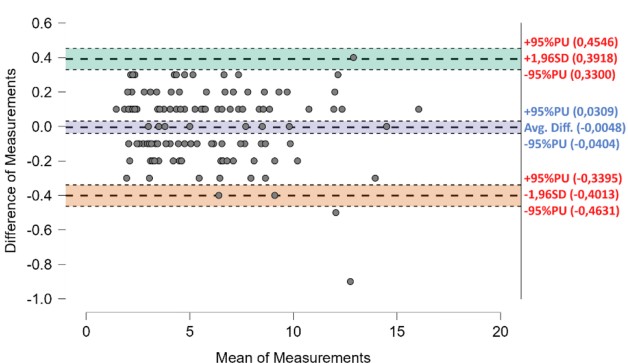

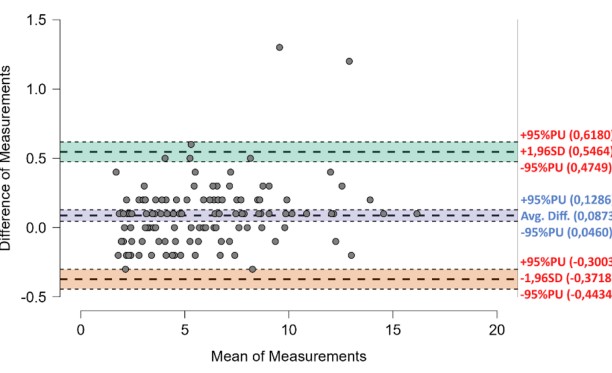

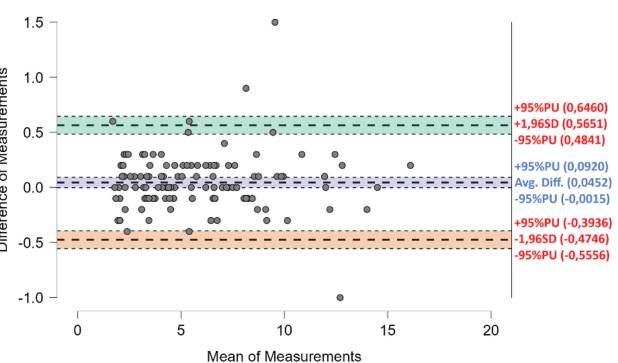

**Figure 3 Bland-Altman plots showing inter-study and inter-investigator agreement for right hand proprioception.**

$p$ = 0.0001). Higher values were recorded by both Rater I and Rater II in Exam I. Nevertheless, ultimately the agreement between the measurements was confirmed by high Pearson's linear correlation coefficients *(r = 1.00)* and ICC (0.994–0.999) (Table 3).

By interpreting the results shown by the Bland-Altman plots for left hand proprioception measurement it was possible to determine that the highest level of agreement existed between Rater I and Rater II in Exam II. Mean deviation was only 0.013° whereas deviations in the specific values did not exceed +0.4°. The highest mean deviations were observed in measurements recorded by Rater II during Exam I and Exam II (0.13°), with differences of nearly 2.5° in isolated cases. Despite the above observation, on the whole it can be said there is agreement between the measurements performed since a large majority of the differences in the results did not exceed 1.0° (Fig. 4).

## DISCUSSION

The aim of the study was to assess the inter-rater and intra-rater agreement of measurements performed using the Luna EMG multifunctional robot as a tool for assessing upper limb proprioception in patients with stroke. The research has shown that Luna EMG is a reliable tool in the evaluation of upper limb proprioception. Measurements made with Luna EMG show high inter-rater and intra-rater reliability of the assessment of the proprioceptive sensation of the upper limb in individuals with stroke.

In the international literature there are no scientific reports investigating this subject matter, *i.e.*, Luna EMG aided assessment of upper limb proprioception in patients with stroke. The only related studies published so far have presented evidence showing the reliability of this tool in healthy populations, when the device was applied to assess the sense of knee position (*Oleksy et al., 2022*) and to measure upper limb proprioception (*Leszczak et al., 2024*). The former study showed high reliability (ICC = 0.866–0.982) of Luna EMG aided assessments of both knee flexion and extension in active and passive modes in healthy individuals (*Oleksy et al., 2022*). Likewise, the latter study demonstrated a high consistency (ICC = 0.969–0.997) of upper limb proprioception measurements performed using the device (*Leszczak et al., 2024*).

In addition, the Cohen's d effect size was calculated. The Cohen's d results show small practical differences in measurements between studies and between researchers. The three highest values of the order of 0.343; 0.359; 0.373 demonstrate a low-moderate effect size. The remainder are at a weak level. Referring to *Ellis (2010)* norms: 0.2–weak effect; 0.5–moderate effect; 0.8–strong effect.

The present study was intended as the next step in investigating the effectiveness of Luna EMG, with a focus on patients with stroke, a neurological condition adversely affecting proprioception. Research shows that approximately 50% of individuals after a stroke experience upper limb proprioception deficits (*He et al., 2022*; *Meyer et al., 2016*), which negatively affects functional performance (*Ingemanson et al., 2019*), activity and participation in daily life (*Carey, Matyas & Baum, 2018*) and motor control (*Carlsson, Gard & Brogårdh, 2018*). The present study was conducted in a group of 126 patients with chronic stroke. The sample size is a strength of the study. The number of participants enrolled is representative, and the sample size was calculated using a statistical research

**Table 3 Agreement between measurements of left hand proprioception.**

**Proprioception (left hand) [°]**

| Exam | Rater | Measurement $\bar{x}$ | SD | Differences $\bar{x}$ | SD | p | r | ICC | CV | SEM |
|------|-------|------|------|------|------|------|------|------|------|------|
| I | 1 | 6.65 | 3.52 | 0.02 | 0.25 | 0.4487 | 0.998 | 0.998 | 52.98 | 0.31 |
| I | 2 | 6.67 | 3.53 | | | ($d = -0.068$) | | (0.997–0.998) | 52.99 | 0.31 |
| II | 1 | 6.55 | 3.43 | −0.01 | 0.18 | 0.4027 | 0.999 | 0.999 | 52.34 | 0.31 |
| II | 2 | 6.54 | 3.45 | | | ($d = -0.024$) | | (0.998–0.999) | 52.82 | 0.31 |
| I | 1 | 6.65 | 3.52 | −0.10 | 0.28 | 0.0002 | 0.997 | 0.996 | 52.98 | 0.31 |
| II | 1 | 6.55 | 3.43 | | | ($d = 0.343$) | | (0.995–0.997) | 52.34 | 0.31 |
| I | 2 | 6.67 | 3.53 | −0.13 | 0.35 | 0.0001 | 0.995 | 0.994 | 52.99 | 0.31 |
| II | 2 | 6.54 | 3.45 | | | ($d = 0.359$) | | (0.992–0.996) | 52.82 | 0.31 |

**Note**:
$\bar{x}$, mean; *SD*, standard deviation; °, degrees of angle; *p*, assessment of the significance of differences in the average level of two series of measurements (t-test for related samples); *d*, wielkość effect size; d, Cohena; *r*, Pearson's linear correlation coefficient; ICC, intraclass correlation coefficient with 95% confidence interval; CV, coefficient of variation; SEM, standard error of the mean.

### Exam I Researcher 1-2

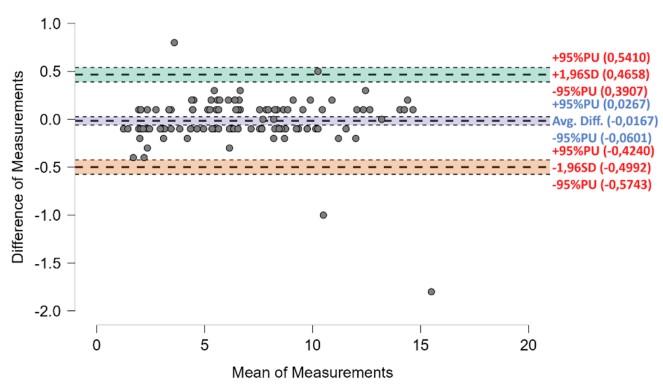

### Exam II Researcher 1-2

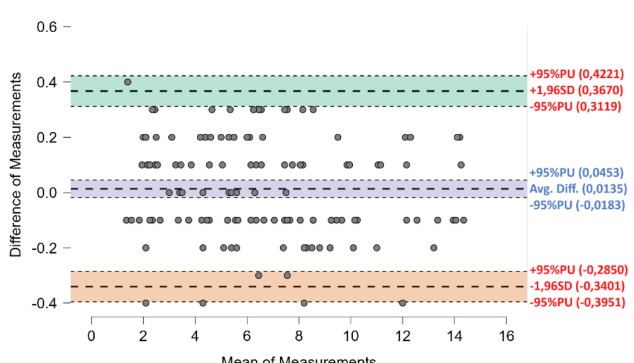

### Exam I-II Researcher 1

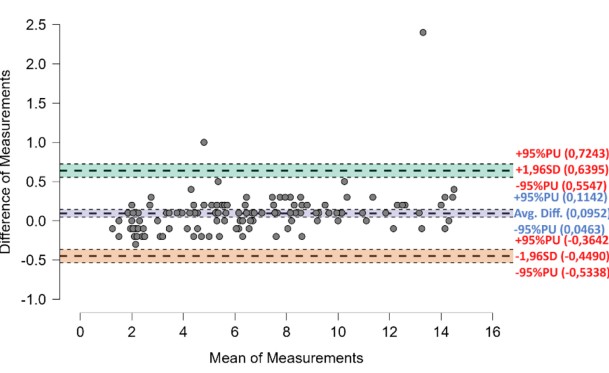

### Exam I-II Researcher 2

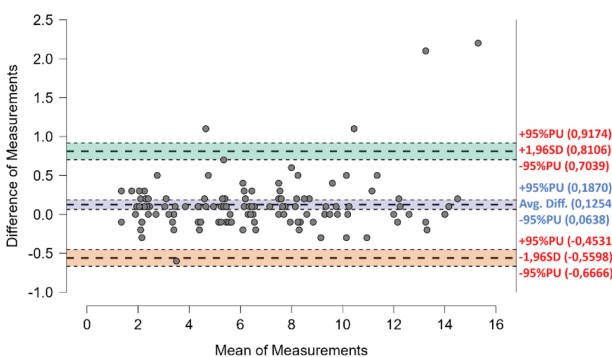

**Figure 4 Bland-Altman plots showing inter-study and inter-investigator agreement for left hand proprioception.**

Leszczak et al. (2024), *PeerJ*, DOI 10.7717/peerj.17903     **10/15**

planning method. The authors decided to focus on patients with chronic stroke because research shows that the most pronounced progress in the recovery of neuromotor function is observed at early stages post-stroke (*Duncan, Lai & Keighley, 2000*; *Lee et al., 2015*) due to the fact that changes associated with regeneration within the penumbra occur relatively quickly after stroke onset and the process slows down in the later stages (*Dąbrowski et al., 2019*; *Kopp et al., 1990*). Therefore, the study took into account patients at a chronic stage post-stroke who may have already adapted to the use of the persistent motor pattern in the affected upper limb and have developed compensatory strategies in the unaffected limb.

Another reason for undertaking this research lies in the fact that very few studies so far have investigated upper limb proprioception in patients with stroke (*Rand, 2018*; *Kiper et al., 2015*; *Ocal, Alaca & Canbora, 2020*); however, the findings reported are rather promising. As an example, *Kiper et al. (2015)* demonstrated that proprioceptive training may lead to improvements in patients with upper limb paralysis after subacute stroke. Furthermore, a study by *Ocal, Alaca & Canbora (2020)* showed that, compared to a conventional therapy, proprioceptive training of the upper limb more effectively increases the frequency and quality of movements performed with the upper limb by patients at a chronic stage post-stroke. Therefore, we believe that it is necessary to continue research focusing on various issues associated with upper limb proprioception post-stroke. Our findings show that the Luna EMG multifunction robotic device is a reliable tool in the evaluation of upper limb proprioception in patients with chronic stroke. It is a well-established fact that diagnostic assessments are essential for proper planning and monitoring of training-based therapies (*Sarzyńska-Długosz, 2023*); therefore, in further studies we intend to evaluate the effectiveness of Luna EMG in assessing the effects of upper limb proprioceptive training in patients after stroke.

In summary of these considerations we can say that this study presents the first scientific evidence related to evaluation of upper limb proprioception in patients with chronic stroke, performed using the Luna EMG multifunction robotic device, and the findings are consistent with results of other studies involving healthy participants, and they demonstrate high reliability of the tool in evaluating upper limb proprioception in patients at a chronic phase after stroke.

### Limitations

The study has some limitations, most importantly, evaluation of upper limb proprioception performed with the Luna EMG diagnostic and rehabilitation device was not supported with results acquired using proprioception assessment scales and tests. However, we are planning further research in which effects of stroke rehabilitation will be evaluated using Luna EMG as well as assessment scales and tests.

## CONCLUSIONS

The study demonstrates that the Luna EMG multifunction robotic device is a reliable tool in the evaluation of upper limb proprioception. Measurements made with Luna EMG show high inter-rater and intra-rater agreement in the assessment of the proprioceptive sensation of the upper limb in patients with stroke.

## ACKNOWLEDGEMENTS

The authors are most grateful to all participants for their committed involvement in the study protocol.

### Funding

The research presented in this article was co-financed by the European Union from the European Regional Development Fund, Smart Growth Operational Programme, grant no. POIR.01.01.01-00-2077/15 "Development of innovative methods of automatic diagnostics and rehabilitation using robots and bioelectric measurements". The funders had no role in study design, data collection and analysis, decision to publish, or preparation of the manuscript.

### Grant Disclosures

The following grant information was disclosed by the authors:
European Regional Development Fund, Smart Growth Operational Programme: POIR.01.01.01-00-2077/15.

### Competing Interests

The sponsor of the study was EGZOTech and this fact may be considered a possible conflict of interest. However, EGZOTech employees (Anna Poświata, Anna Roksela, Michał Mikulski) were not involved in direct activities related to the study protocol (patient recruitment, data collection, analysis of results), but supported the researchers by conducting consultations on the robotic technology that EGZOTech developed as a company and provided the equipment necessary for this study. Additionally, these authors: participated in designing the study, conducting a literature review, establishing inclusion and exclusion criteria, prepared the tables, writing the research methodology and accepting the manuscript.

### Author Contributions

- Justyna Leszczak conceived and designed the experiments, performed the experiments, analyzed the data, prepared figures and/or tables, authored or reviewed drafts of the article, and approved the final draft.
- Bogumiła Pniak performed the experiments, prepared figures and/or tables, and approved the final draft.
- Mariusz Drużbicki analyzed the data, authored or reviewed drafts of the article, and approved the final draft.
- Anna Poświata conceived and designed the experiments, prepared figures and/or tables, authored or reviewed drafts of the article, and approved the final draft.
- Michał Mikulski conceived and designed the experiments, prepared figures and/or tables, authored or reviewed drafts of the article, and approved the final draft.

- Anna Roksela conceived and designed the experiments, prepared figures and/or tables, and approved the final draft.
- Agnieszka Guzik conceived and designed the experiments, performed the experiments, analyzed the data, authored or reviewed drafts of the article, and approved the final draft.

### Human Ethics

The following information was supplied relating to ethical approvals (*i.e.*, approving body and any reference numbers):

The University of Rzeszow (resolution no. 2022/036/W) approved the study.

### Data Availability

Data are available in the Supplemental Files.

### Supplemental Information

Supplemental information for this article can be found online at http://dx.doi.org/10.7717/peerj.17903#supplemental-information.

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
