# Peer review of "Assessment of inter-rater and intra-rater reliability of the Luna EMG robot as a tool for assessing upper limb proprioception in patients with stroke—a prospective observational study"

_PeerJ, doi:10.7717/peerj.17903_

## Round 0.1 · original submission · Minor Revisions

Please read the comments from both peer reviewers and revise your manuscript.

Reviewer 1 ·

Basic reporting

This is an interesting article that focus on reliability of LUNA EMG robot in stroke population. In general, the sentences were fair and may benefit from proof reading service to minimize grammar errors.


Abstract.
-The abstract already discuss comprehensively on your paper, although I might say some of the issues can be improved, for example “on average aged nearly 60 years”, can be replaced by your true study population data.

Experimental design

Introduction.

1) The introduction was well written
2) The research gap is well explained and the usage of outcome measures are appropriate for this study.

Methodology

1) Which sample size calculator that you used? (website url?, application version?)
2) Vague inclusion criteria. What is completed first ischemic stroke? How about othern europathy that can affect propioception?


Discussion & conclusion.

1) Interprets the findings of this study with good comparison with previous studies in this topic.
2) Good limitation and recommendation

Conclusion
1) Good conclusion

Validity of the findings

All the outcome measures used support the study objectives.

Additional comments

This can be a highly cited paper for researchers using the Luna EMG in the future.

Reviewer 2 ·

Basic reporting

This study provides a robust background on the importance of proprioception and its assessment, especially post-stroke, effectively situating the study within existing research. This is a well-structured academic paper. However, the figures need some further improvement in the resolution and labeling. Several areas in the formatting and detail of reporting need further attention to meet the publication standards. Below are some detailed comments:

Line 10: Please spell out EMG when you first mention the word.

Lines 40-45: This section effectively highlights the impact of impaired proprioception on stroke patients. Perhaps elaborate on the typical outcomes or statistics that underline the severity of proprioception loss in stroke rehabilitation, to contextualize the study's importance.

Line 97-98: The phrase "unstable medical condition" is somewhat vague; consider specifying what constitutes an 'unstable' condition, or provide examples.

Line 106-114: Please improve the writing of the paragraph.

Line 149-159: Please include what statistical software you used to conduct the statistical analysis.

Lines 163-168: There's a mention of significant differences in measurements between raters and exams with p-values reported (p=0.0001 and p=0.0220). However, there's no discussion on the effect size or the practical significance of these differences. To improve the statistical report, you can include the mean differences to provide context on the magnitude of these differences. This addition will help in understanding the practical implications of the observed statistical significance.

Line 294: There seems to be an issue with the indent.

Figure 1:
1. there seems to be a typo "Folow diagram".
2. It seems that only patients who completed Exam I (N=130) qualified the Exam II. If that's the case, you should make the Exam II below Exam I to avoid confusion.
3. Please use the same word consistently, "patients", "Persons", "individuals"

Figure 3
1. it would be beneficial to ensure that all figures are of high resolution as per the journal's guidelines.
2. Please add a label to the figure (mean, +1.96SD, -1.96SD). I understand that you have three dashed lines in the figure, the upper, middle, and lower dashed line stands for +1.96Sd, -1.96SD. However, it is not clear why there are green, purple, and orange areas around the dashed lines.

Table 1 and Table 2
You could combine Table 1 and Table 2 for simplicity when reporting demographic variables. In my opinion, it seems redundant to report the two tables separately. Also, please consider reporting other relevant variables that might help readers to have a better understanding of the demographics and medical history of the patients.

Table 3 and Table 4
1. You did not mention what unit was those measurements in. Are they in degrees of angle? Please state them in the Tables and the manuscript.
2. You have reported perfect Person’s correlation coefficient(r=1); it would be more informative if you could report them in three digits. On the column name, it would be better if you could replace “R” with “r”.

Experimental design

The original primary research is within the aim of the scope of the journal. The research question is well-defined, relevant, and meaningful. The methods described are with sufficient detail and information to replicate. Here are some detailed comments:

Lines 92-94: The inclusion and exclusion criteria are well-defined. It would be helpful if you consider discussing why specific criteria like “subsequent stroke”, “haemorrhagic stroke”, and “stroke of the brainstem” were excluded, providing a rationale based on the study’s scientific goals or literature.

Line 121: Please clarify under what conditions two independent raters conducted the assessments, and how long the assessments took.

Line 127-128: The description of the Luna EMG's use is clear. Adding information about the calibration process or settings used could help other researchers replicate the study conditions more precisely.

Validity of the findings

The statistical methods used are appropriate for the study design, and the results are reported with clarity. The high interclass correlation coefficients indicate robust data. Here are some detailed comments:

Line 164-168: The manuscript notes significant differences in measurements between raters and exams. Clarifying whether these differences impact the study's conclusions about the Luna EMG's reliability would be helpful.

Lines 169-171: Pearson's linear correlation and ICC values are impressively high. It might be beneficial to discuss any potential biases or factors that could have influenced these high correlations to ensure the robustness of the results.

Additional comments

The manuscript is a valuable contribution to the field of rehabilitation robotics and stroke recovery. It is recommended for publication after minor revisions.

---

## Round 0.2 · accepted · Accept

Thank you for your revised manuscript which has been accepted.

Reviewer 2 ·

Basic reporting

No comment. The authors have addressed all my previous comments.

Experimental design

No comment. The authors have addressed all my previous comments.

Validity of the findings

No comment. The authors have addressed all my previous comments.

Additional comments

The revision of "Assessment of Inter-rater and Intra-rater Reliability of the Luna EMG Robot as a Tool for Assessing Upper Limb Proprioception in Patients with Stroke - A Prospective Observational Study" have addressed all my previous comments. The authors have made the necessary revisions, ensuring clarity, thoroughness, and scientific rigor in their study. The methodology, data analysis, and discussion have been comprehensively enhanced, providing a robust assessment of the reliability of the Luna EMG Robot.